# Machine Translation for Low-resource Finno-Ugric Languages

**Lisa Yankovskaya   Maali Tars   Andre Tättar   Mark Fishel**
Institute of Computer Science
University of Tartu, Estonia
{lisa.yankovskaya,maali.tars,andre.tattar,mark.fishel}@ut.ee

## Abstract

This paper focuses on neural machine translation (NMT) for low-resource Finno-Ugric languages. Our contributions are three-fold: (1) we extend existing and collect new parallel and monolingual corpora for 20 Finno-Ugric languages, (2) we expand the 200-language translation benchmark FLORES-200 with manual translations into nine new languages, and (3) we present experiments using the collected data to create NMT systems for the included languages and investigate the impact of back-translation data on the NMT performance for low-resource languages. Experimental results show that carefully selected back-translation directions in a multilingual setting yield the best results in terms of translation scores, for both high-resource and low-resource output languages.

## 1   Introduction

Neural networks have caused rapid growth in output quality for many natural language processing tasks, including neural machine translation (NMT, Vaswani et al., 2017). However, the output quality crucially depends on the availability of large amounts of parallel and monolingual data for the covered languages.

Recently synthetic data and cross-lingual transfer have not only shown potential for low-resource language NMT but also have been taken to the extreme through open massively multilingual translation models (Fan et al., 2021; NLLB Team et al., 2022). In addition to translation models, a massive translation benchmark FLORES-200 (NLLB Team et al., 2022) has been created, consisting of multi-parallel translations of the same sentences into 200 languages.

Here we focus on NMT for low-resource languages from a family of languages spoken in Europe, but not part of the Indo-European family: Finno-Ugric languages. Three members of that family (Estonian, Finnish and Hungarian) are commonly included in massively multilingual efforts and can be considered medium-resource languages. At the same time, several lower-resource Finno-Ugric languages are not included in the existing massively multilingual models (M2M-100, NLLB). In terms of the number of speakers, they range from 20 near-native speakers of Livonian to several hundred thousand speakers of Mordvinic languages.

Our contributions are three-fold. First, we present a collection of parallel and monolingual corpora that can be used for training NMT systems for 20 low-resource Finno-Ugric languages. The resources are collected from sources that are already digital (primarily online sources); the languages and the data are described in Section 3.

Secondly, we expand a part of the 200-language translation benchmark FLORES-200 with manual translations into the low-resource Finno-Ugric languages. This includes the first 250 sentences of FLORES-200 and the following languages: Komi, Udmurt, Hill and Meadow Mari, Erzya, Livonian, Mansi, Moksha and Livvi Karelian. This new benchmark is described in Section 4.

Finally, we use the collected parallel and monolingual data in experiments to create NMT systems for the covered languages. The main question we address is which subsets of translation directions yield the best results for the included low-resource languages. We achieve an average chrF++ score of 26.8 when translating from high-resource to low-resource languages included in our expansion of FLORES-200. The complete experiments and results are presented in Sections 5 and 6.

## 2  Related Work

**Low-resource NMT**  Machine translation is dominated by neural methods in current research. Neural machine translation also requires large amounts of training segments for high-quality translation across different domains. That is a challenge when it comes to low-resource languages.

In Gu et al. (2018) and Sennrich and Zhang (2019), the authors investigate the best NMT model setups, with Sennrich and Zhang (2019) showing a comparison to phrase-based systems that are not that common these days. Gu et al. (2018) and Kocmi and Bojar (2018) indicate that training universal models (sharing parameters between multiple languages) and transfer learning are two aspects that get significant gains for low-resource language pairs in translation quality.

More recently, low-resource machine translation has risen to the attention of more and more research groups with multiple comprehensive surveys emerging (Haddow et al., 2022; Wang et al., 2021), showing that there has already been a lot of work done that can now be systematically aggregated and utilized in further research.

**Low-resource NMT for Finno-Ugric languages**  Some of the Finno-Ugric languages have been considered in the context of NMT before. Tars et al. (2021, 2022a,b) and Rikters et al. (2022) present experiments with several Sami languages, Võro and Livonian. They used similar techniques like multilinguality, pre-trained models, transfer learning, and back-translation to better the translation quality. Our work aims to bridge the gap between the other low-resource Finno-Ugric languages and those that already have good support, offered by the previously published papers.

In 2022, Livonian-English was part of the translation shared task at WMT, the International Conference of Machine Translation (Kocmi et al., 2022). A Livonian-English test set was created; in our work, we add Livonian to FLORES-200, which covers several language pairs more than the WMT'22 test set.

**Back-translation in low-resource setting**  Back-translation is a widely used method for enhancing translation quality while making use of monolingual data (Sennrich et al., 2016). This is also one of the aspects that allows for good quality NMT systems in the low-resource setting

because low-resource languages lack parallel data while monolingual data is often much easier to find.

There has been research into exploring the specifics of back-translation like models used for synthetic data creation, beam search vs greedy search, the domain of monolingual data as well as amounts of synthetic data (Edunov et al., 2018), the last of them is the closest we also desire to investigate in our low-resource Finno-Ugric setting. Other research goes into detail about how diverse the synthetic data should be (Burchell et al., 2022) and how effective iterative back-translation is (Hoang et al., 2018).

**Pre-trained models**  For multilingual NMT, it has become insufficient to train models from scratch, instead using pre-trained models has become a prevalent method for all NLP tasks. In machine translation, the massively multilingual models of M2M-100 and NLLB are a good starting point to use for fine-tuning and transfer learning (Fan et al., 2021; NLLB Team et al., 2022).

## 3  Languages and Data

The Finno-Ugric language group has two major branches: Finno-Permic and Ugric. Although both branches share common linguistic roots, they are quite distant.

The Finno-Permic branch includes two high-resource languages, Estonian and Finnish, and several low-resource languages, such as Komi, Komi Permyak, Udmurt, Hill and Meadow Mari, Erzya and Moksha, Proper and Livvi Karelian, Ludian, Võro, Veps, Livonian, Sami languages. The Ugric branch comprises three languages: high-resource Hungarian and two low-resource Mansi and Khanty.

In this work, we develop an NMT system between 20 low-resource Finno-Ugric (FU) languages shown in Figure 1 and seven high-resource languages (English, Estonian, Finnish, Hungarian, Latvian, Norwegian (Bokmål), and Russian). The selection of the high-resource languages is not accidental: Estonian, Finnish, and Hungarian belong to the FU language family, while Latvian has markedly influenced Livonian, Norwegian has deeply affected the Sami languages and Russian has had a profound impact on the Permic, Mordvinic, Mari, Karelian, Veps, and Ob-Ugric languages.

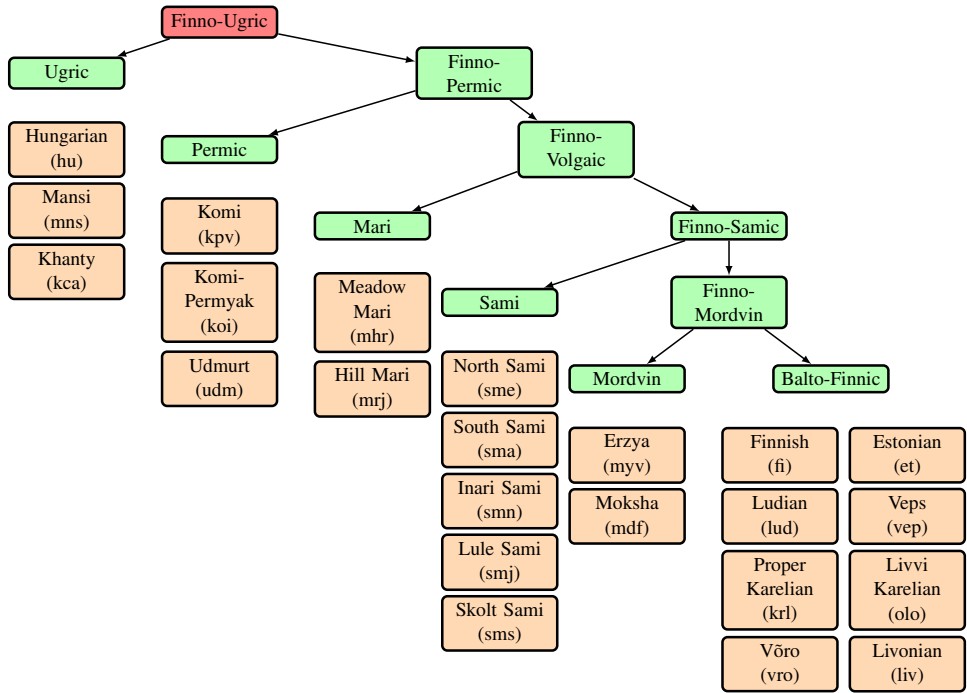

Figure 1: Languages from the Finno-Ugric language family, for which we have created MT systems. Green colour represents branches, orange — languages. The Finno-Permic languages are visualized according to the Janhunen classification (Janhunen, 2009).

### 3.1 Monolingual corpora

We collected monolingual corpora mainly by crawling texts off the web and combining with pre-existing corpora. Three main categories of texts can be distinguished: news, Wikipedia, and biblical. Texts that do not fall into these categories have been grouped together under the category "Other". Table 1 provides more information of the amount of data collected.

Wikipedia texts were collected from the Wortshatz corpora collection (Goldhahn et al., 2012) and the Tatoeba Translation Challenge corpora (Tiedemann, 2020).

The biblical subcorpus consists of texts taken from the Finugorbib[1] and the open corpus of Veps and Karelian languages "VepKar" (Boyko et al., 2022)[2].

In order to create a subcorpus of news, we used the following online news media:

- **Komi (kpv)**: http://komikerka.ru/, https://komiinform.ru/news/e/161, https://www.nbrkomi.ru/kraevedenie/vyltoryas

- **Udmurt (udm)**: https://udmddn.ru/ivorjos/, https://oshmes.info/

- **Erzya (myv)**: https://vk.com/club78443596

- **Moksha (mdf)**: https://mokshapr.ru/

- **Livvi Karelian (olo)**: https://www.omamedia.ru/ka/

- **Veps (vep)**: https://www.omamedia.ru/ve/

- **Mansi (mns)**: https://khanty-yasang.ru/

- **Khanty (kca)**: https://khanty-yasang.ru

The subcorpus "Other" is a collection of texts from the Mozilla dataset of voices "Common Voice"[3] and the open corpus of Veps and Karelian languages "VepKar".

Monolingual data for most of the high-resource languages (English, Estonian, Finnish, Hungarian, Latvian, Russian) was sampled from the WMT news dataset[4]. The Norwegian monolingual data was sampled from the "Norsk aviskorpus"[5]. Parallel data between high-resource languages was sampled from OPUS (Tiedemann, 2012).

We share the part of the monolingual corpora[6].

---

[1]http://www.finugorbib.com/alt/alt_al.html
[2]http://dictorpus.krc.karelia.ru/en

[3]https://commonvoice.mozilla.org/en/datasets
[4]https://data.statmt.org/news-crawl/
[5]https://www.nb.no/sprakbanken/ressurskatalog/oai-nb-no-sbr-4/
[6]https://huggingface.co/datasets/tartuNLP/smugri-data

| | mono | | | | |
|---|---|---|---|---|---|
| | wiki | bible | news | others | total |
| kpv | 18.4 | 4.5 | 38.3 | | 61.2 |
| koi | 11.5 | 1.2 | | | 12.7 |
| udm | 43.5 | 3.7 | 36 | | 83.2 |
| mrj | 49.5 | | | 14.6 | 64.1 |
| mhr | 141 | | | 109 | 251 |
| myv | 73.8 | | 1.3 | 7.7 | 82.8 |
| mdf | 8 | 3.9 | 3.9 | 0.3 | 16.1 |
| krl | | 1.8 | | 18.4 | 20.2 |
| lud | | | | 5.3 | 5.3 |
| olo | | | 21 | 19.4 | 40.4 |
| vep | 71.3 | 0.9 | 7.8 | 35.3 | 115.3 |
| vro | | | | 162 | 162 |
| liv | | | | 40 | 40 |
| sma | | | | 55 | 55 |
| sme | | | | 34 | 34 |
| smj | | | | 128 | 128 |
| smn | | | | 123 | 123 |
| sms | | | | 76.7 | 76.7 |
| mns | | 0.8 | 10.3 | | 11.1 |
| kca | | 0.8 | 13.3 | | 14.1 |

Table 1: The collected monolingual corpus of the low-resource languages. The figures in the table are in thousands of sentences.

### 3.2 Parallel Corpora with Russian

As the majority of speakers of the low-resource FU languages live in Russia, most of the parallel translations we have collected are in Russian (see Table 2). A substantial portion of the parallel corpus consists of biblical texts from the Finugorbib and "VepKar". The rest of the parallel corpus comprises various texts, mostly collected from the "VepKar" and Finnougoria webpage[7].

### 3.3 Data for Võro, Livonian, and Sami Languages

The data (parallel and monolingual) for the Võro, Livonian, and Sami languages that we included in our experiments were taken from the previous editions of NMT developments with low-resource FU languages (Tars et al., 2021, 2022b; Rikters et al., 2022). Võro data is mostly from a META-

[7]https://finnougoria.ru/

| | parallel (Ru) | | |
|---|---|---|---|
| | bible | others | total |
| kpv | 11 | 2 | 13 |
| koi | 8 | 0.3 | 8.3 |
| udm | 30 | | 30 |
| mrj | 8 | | 8 |
| mhr | 9 | | 9 |
| myv | 11.5 | 0.9 | 12.4 |
| mdf | 11.5 | 1 | 12.5 |
| krl | 10.5 | 7.7 | 18.2 |
| lud | | 10.5 | 10.5 |
| olo | 11.9 | 4 | 15.9 |
| vep | 16.4 | 11.1 | 27.5 |
| mns | 0.7 | | 0.7 |
| kca | 2 | | 2 |

Table 2: The collected parallel corpus with Russian. The figures in the table are in thousands of sentences.

SHARE[8] source consisting of newspapers, fiction, and a handful of other domains. Livonian data comes from OPUS (Tiedemann, 2012) Liv4ever dataset. Sami language data was collected in previous works from the resources of The Arctic University of Norway[9].

## 4 Benchmark dataset

In order to create a multilingual benchmark for Finno-Ugric languages[10], we took the first 250 rows of the FLORES dataset (NLLB Team et al., 2022) and had them translated into nine Finno-Ugric languages: Komi, Udmurt, Hill and Meadow Mari, Erzya, Moksha, Livonian, Mansi, and Livvi Karelian by a team of bilingual speakers, both natives and fluent speakers, of Estonian or Russian and low-resource FU languages. Most translators have an academic degree in linguistics or have extensive translation experience.

While translating, translators have encountered the following problems:

1) Some sentences of the FLORES dataset contain very specific vocabulary, such as "barbs" or "barbules", which can be hard to translate because

[8]https://doi.org/10.15155/1-00-0000-0000-0000-001A0L
[9]https://giellalt.uit.no/tm/TranslationMemory.html
[10]https://huggingface.co/datasets/tartuNLP/smugri-flores-testset

the translators are unfamiliar with this scientific domain.

2) Some words, such as "inning" or "shuttle", are not commonly used or have never been used in some FU languages. As a result, translators have had to create new words based on their sense of the language.

3) The FLORES dataset contains a few lengthy sentences, whereas, in some FU languages, it is preferable to use shorter sentences. So the long sentences have been divided into shorter sentences

While working on creating new benchmark datasets, we found a broken row in the original English dataset: "Singer Sanju Sharma started the evening, followed by Jai Shankar Choudhary. *esented the chhappan bhog bhajan as well.* Singer, Raju Khandelwal was accompanying him." As we can see, the second sentence makes no sense. To fix it, (i) we have omitted this sentence in the English, Latvian, Norwegian (Bokmål), and Russian datasets; (ii) we have added the translation of the last sentence, which was missing, to the Estonian dataset ("Õhtut alustas laulja Sanju Sharma, kellele järgnes Jai Shankar Choudhary. Laulja Raju Khandelwal oli teda saatmas"); (iii) we have edited the first sentence in the Finnish dataset by removing part of it ("Illan aloitti laulaja Sanju Sharma, ja häntä seurasi Jai Shankar Choudhary ~~, joka esitti myös chhappan bhogien bhajanin~~. Häntä säesti laulaja Raju Khandelwal."); (iv) we have replaced the second sentence in the Hungarian dataset ("Sanju Sharma énekes indította az estét, õt követte Jai Shankar Choudhary. ~~pedig a chhappan bhog bhajant adta elő Raju Khandelwal kíséretében.~~ Raju Khandelwal énekes kísérte.").

# 5 Experiments

One of the goals of our paper was to find out which language pairs are needed to reach a certain level of quality for low-resource NMT models in the Finno-Ugric setting. More specifically, the question is whether it is necessary for low-resource multilingual systems to back-translate in all directions (which is costly) or subsets of translation directions can suffice? By finding optimal amounts of synthetic data we can optimize the overall system creation process by making it less costly and less time-consuming while being able to increase the number of iterations performed.

## 5.1 Experiment setup

The baseline in this work is a pre-trained multilingual neural machine translation model (M2M-100, 1.2 billion parameters) that has been fine-tuned on parallel data of previously unseen language pairs in addition to sampled high-resource language pairs to reduce catastrophic forgetting (20k samples per high-resource language pair).

For the back-translation experiments, we designed four sets of back-translation data:

1. Synthetic data between all languages (702 language pairs) (`all-all`).

2. 10% of synthetic data of every language pair in the first set (`all-all-10`).

3. Synthetic data from each low-resource language to each high-resource language and vice versa (for example Udmurt-English, Estonian, Finnish, Latvian, Norwegian, Hungarian, Russian) (`L-H`).

4. Synthetic data from each low-resource language to its related high-resource languages and languages it had original parallel data with and vice versa (for example Udmurt-Estonian, Finnish, Russian) (`L-rH`).

All of the sets had an upper limit of 100k synthetic segments per language pair.

The third and fourth sets were chosen a bit more strategically, incorporating linguistic knowledge about the low-resource languages. The third set was created to see whether high-resource monolingual data helps the low-resource languages more efficiently when we do not have other data distracting the model. The fourth set included synthetic data for each low-resource language to its related high-resource languages plus language pairs that it already had parallel data with.

## 5.2 Technical specifications

We trained all the described NMT systems on the LUMI[11] supercomputer. All models were fine-tuned with the Fairseq framework (Ott et al., 2019) implementation of M2M-100 (Fan et al., 2021) for 350k updates with a batch size of 3840 tokens (the number was chosen to match earlier versions of models trained with the Huggingface implementation of M2M-100). All models were fine-tuned on 4 AMD Mi250X GPU-s. We used custom

---

[11]https://www.lumi-supercomputer.eu/about-lumi/

scripts[12] to expand the embedding matrix and the vocabulary of M2M-100.

# 6 Results

**Quantitative analysis** To get an overview of the quality of the models and compare different synthetic data settings, we compare chrF++ (Popović, 2015, 2017) results for all of the experiments, calculated using sacreBLEU (Post, 2018)[13]. As we are evaluating morphologically rich languages, reporting chrF++ as the main automatic metric gives the most truthful results, whereas BLEU (Papineni et al., 2002) is too punishing on this type of languages.

In Table 3, we display comparisons of all five models (baseline and four models with different synthetic datasets) with different clusterings of language pairs.

In the subtable 3a, we notice that adding synthetic data from every language pair damages the translation quality translating into low-resource languages. Comparing `all-all` and `all-all-10` models, where `all-all-10` contains 90% less synthetic data, higher quality is obtained by the `all-all-10` over all of the language pairs as well as translating into low-resource languages. This means that better results are achieved with less synthetic data and less training time/resources used.

The cause of this situation is the fact that although we limited monolingual data to 100k for each language pair, some smaller language pairs had a lot less than 100k monolingual sentences. Taking only 10% of the synthetic data leveled the distribution of high- and low-resource synthetic data and allowed high-resource to low-resource pairs to get more attention during training.

The best scenario for translating into low-resource languages seems to be to use synthetic data from low-resource language into related high-resource languages (`L-rH`). This is shown by the subtables 3a and 3c. For translating from low-resource languages to high-resource languages, however, the most efficient is to add synthetic data from each low-resource language to all the high-resource languages involved in the initial fine-tuning (`L-H`), instead of using the larger `all-all` model.

---

[12]https://github.com/TartuNLP/m2m-100-finetune
[13]sacreBLEU signature:
```
nrefs:1|case:mixed|eff:yes|
nc:6|nw:2|space:no|version:2.0.0
```

Comparing the baseline to all the other models, we see significant improvements which can be explained by the fact that the parallel data for low-resource languages originated mainly from the bible, but monolingual data originated from different domains, even more for the high-resource languages.

One anomaly clear from subtable 3c, is the Mansi language performing badly across all of the models with the highest score being 10+ points below the scores for other languages. After further inspection, the fault seemed to be the non-normalized symbols in the dataset which were not included in the dictionary before training and were causing unknown symbols in the translations.

We do not report results for low-resource languages that lack the FLORES benchmark, because the held-out test set is too biased towards the bible domain and there is no other comprehensive benchmark for the rest of the low-resource languages.

In addition to the mentioned experiments, we tried filtering the back-translation data with some of the same filters used to filter the original parallel data. However, the results of the experiments with filtered back-translation data were the same or even a little worse than with the non-filtered back-translation data. Thus, we do not report these results and leave the thorough back-translation filtering analysis for future work.

**Comparison to previous results** To compare some of the language pairs to previous results on already existing test sets (Tars et al., 2022b,a), we offer a detailed overview of high- to low-resource translation directions for Võro, Livonian, and all the included Sami languages in Table 4. The improvement with our model varies between the language pairs, but the majority of the compared directions achieve a noticeable gain in BLEU, some even very significant 10 and 20 BLEU point increases which might indicate some test data leakage into the training set. The improvements for English-Livonian are noteworthy because although our model gains only about 0.5 BLEU points, it was trained with fewer back-translation iterations and did not need extra finetuning to the specific language pair. For other translation directions, it can be hypothesized that the improved scores are a result of adding synthetic data because the methods we are comparing to omitted using back-translation.

|          | low-low | low-high | high-low | low-high(rel) | high(rel)-low | all pairs |
|----------|---------|----------|----------|---------------|---------------|-----------|
| `baseline` | 18.7 | 24.0 | 20.7 | 24.8 | 22.1 | 23.7 |
| `all-all`   | 20.2 | 36.5 | 19.1 | 36.8 | 20.0 | 28.5 |
| `all-all-10` | 25.9 | 34.3 | 24.1 | 34.9 | 25.5 | 30.3 |
| `L-H`       | 26.6 | **36.6** | 25.8 | **37.0** | 27.2 | **32.3** |
| `L-rH`      | **27.2** | 35.5 | **26.8** | 36.1 | **28.2** | 32.0 |

(a) low - low-resource languages, high - high-resource languages, rel - related languages to respective low-resource language. "-" indicates two-way translation directions between the languages.

|          | to-RU | to-EN | to-ET | to-FI | to-HU | to-LV | to-NO |
|----------|-------|-------|-------|-------|-------|-------|-------|
| `baseline` | 19.6 | 25.6 | 26.6 | 25.1 | 22.0 | 24.3 | 24.8 |
| `all-all`   | 42.3 | 39.8 | **28.2** | 36.8 | 35.2 | 37.6 | **35.4** |
| `all-all-10` | 39.2 | 37.5 | 27.8 | 34.7 | 32.4 | 35.2 | 33.6 |
| `L-H`       | **42.9** | **40.4** | 27.8 | **37.2** | **35.5** | **38.0** | 34.6 |
| `L-rH`      | 41.8 | 39.4 | 26.7 | 36.4 | 33.8 | 35.9 | 34.6 |

(b) to-* indicates translation directions from low-resource languages to the respective high-resource language.

|          | to-KPV | to-LIV | to-MDF | to-MHR | to-MNS | to-MRJ | to-MYV | to-OLO | to-UDM |
|----------|--------|--------|--------|--------|--------|--------|--------|--------|--------|
| `baseline` | 15.9 | 28.4 | 22.1 | 21.3 | 12.2 | 19.9 | 22.9 | 22.7 | 21.0 |
| `all-all`   | 15.9 | 26.0 | 18.2 | 24.4 | 12.4 | 15.2 | 16.7 | 21.2 | 21.5 |
| `all-all-10` | 22.3 | 28.6 | 25.2 | 28.3 | 13.7 | 22.1 | 23.1 | 25.3 | 28.1 |
| `L-H`       | 24.6 | 29.5 | 27.0 | **30.8** | 14.3 | 23.8 | 24.4 | **27.1** | 31.2 |
| `L-rH`      | **26.4** | **29.7** | **28.5** | 30.6 | **16.1** | **26.2** | **25.2** | 26.7 | **31.6** |

(c) to-* indicates translation directions from high-resource languages to the respective low-resource language.

Table 3: Average chrF++ results for all experiments across different language pair clusters on FLORES benchmarks. **Bold** - highest score per grouping. `all-all` - contains BT data from every language pair. `all-all-10` - contains 10% of BT data used in `all-all`. `L-H` - contains BT data from each low-resource language to each high-resource language and vice versa. `L-rH` - contains BT data from each language to its related high-resource language + high-resource languages it had parallel data with and vice versa.

|          | en-liv | et-liv | et-vro | fi-sma | fi-sme | fi-smn | fi-sms | no-sma | no-sme | no-smj |
|----------|--------|--------|--------|--------|--------|--------|--------|--------|--------|--------|
| L-rH     | **15.74** | **24.17** | 30.63 | **46.58** | 38.27 | **67.34** | **44.13** | **60.79** | 35.21 | **51.95** |
| *previous best* | 15.19 | 14.51 | **34.11** | 26.63 | **42.89** | 53.3 | 33.72 | 46.79 | **35.38** | 40.01 |

Table 4: BLEU scores for high-resource to selected low-resource languages to compare with previous results in these language pairs. The previous best results are from Tars et al. (2022b,a). The test set is same as used in the previously mentioned publications. **Bold** - best result between our best high-low model and the previous best result.

**Qualitative analysis** Here, we go over the key findings of the qualitative analysis we conducted. We focus and showcase our results on the Komi to Russian translation direction and compare our baseline model and the model trained on back-translated data. The baseline model performed poorly with unnatural and "biblical-looking" translations which is a style introduced by the parallel training data used for the baseline. The baseline model output sounds like Church Slavic, which is a Slavic liturgical language used by the Eastern Orthodox Church, examples of this are both in Figures 2 and 3. The baseline model also introduces biblical artifacts into the translation, which is showcased by an example shown in Figure 2, where "Daesh" is changed

| | |
|---|---|
| Original (kpv) | Полиция юӧртіс, мый уськӧдчӧмын найӧ мыжалӧны чайтана боевикӧс Даешысь (ИГИЛ). |
| Baseline (ru) | Полиция объявила, что его обвиняют в убийстве в Иерусалиме. |
| BT (ru) | Полиция сообщила, что в нападении они подозревают предполагаемого боевика группировки "Даеп" (ИГИЛ) |
| Reference (ru) | Полиция заявила, что в совершении нападения подозревается предполагаемый боевик ДАИШ (ИГИЛ). |
| Reference (en) | Police said they suspect an alleged Daesh (ISIL) militant of responsibility for the attack. |

Figure 2: Example of translations from Komi to Russian. The `Baseline` translation is partially correct. We highlight the word "Jerusalem" in red as it is an artifact (hallucination originating from the Bible) created by the model. The `BT` translation is generally correct, with a small error in the word Daesh, which is highlighted in green. BT refers to the back-translation model, specifically the L-rH model.

| | |
|---|---|
| Original(kpv) | Ӧти воӧн висьысь морт матысса йитчигӧн вермӧ висьмӧдны 10-сянь 15 мортӧдз. |
| Baseline (ru) | И если гонщик вошел в дом ближнего своего, то есть человек, лежащий в язве или в болезни, то есть около десятого, или около пятнадцатого; |
| BT (ru) | В течение одного года риск заражения человека при близких контактах может возрастать с 10 до 15 человек. |
| Reference (ru) | За один год инфицированный человек может заразить от 10 до 15 человек при близком контакте. |
| Reference (en) | In one year's time, an infected person may infect 10 to 15 close contacts. |

Figure 3: Example of translations from Komi to Russian. The translation by the `Baseline` model is generally incorrect, and it is written in the biblical style. The words that stand out as biblical are highlighted in red. The `BT` translation is completely correct. BT refers to the back-translation model, specifically the L-rH model.

into Jerusalem. We found multiple occurrences of Jerusalem in the baseline translations but none of such occurrences in the translations made by the model with the additional back-translated data. Our proposed model, which added a lot of synthetic data into training data, produces much better translations — we hypothesize that this is due to better distribution of data sources, the translations look more general and have an informative style. We also did not notice any named-entity hallucinations. Our findings highlight the importance of data source (domain) and quality in the low-resource scenario, where imbalanced data sources can lead to non-optimal translations.

## 7 Conclusion

We presented a FLORES-based benchmark dataset for nine low-resource Finno-Ugric languages: Erzya, Komi, Livvi Karelian, Livonian, Hill and Meadow Mari, Mansi, Moksha, and Udmurt. In this study, we trained and evaluated multiple models for these languages and generated a large amount of synthetic parallel data through back-translation. The results showed that the models achieved promising performance on the benchmark dataset and demonstrated the potential of these methods for low-resource machine translation. Our experiments also showed that it could be useful to choose back-translation settings more strategically, selecting certain language pairs, to achieve better results while using fewer resources

for back-translation and training.

## Limitations

The machine translation systems described in this paper have several limitations that are important to consider.

- Most of the parallel training data comes from the Bible - this limits the generalizability of the system, for example when trying to translate non-religious texts from Wikipedia.

- Train-Test mismatch, specifically for the parallel training data, impacts the overall trustworthiness of the quantitative results.

- Limited test data coming from a single source - We managed to translate only a quarter of the multilingual FLORES dataset. Also, we only have the FLORES dataset which originates from [English] Wikipedia.

- Finno-Ugric languages written in the Cyrillic alphabet might benefit from transliteration, which we did not try in this study. Transliteration converts text written in one script into another script. It remains an open question if transliteration into the Latin script would improve the translation quality.

These limitations highlight the need for further research in machine translation for Finno-Ugric languages. Future studies should address these limitations.

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
