# OpenReview forum: "Machine Translation for Low-resource Finno-Ugric Languages"
_NoDaLiDa/2023/Conference — NoDaLiDa 2023_

### Official Review · Reviewer_nWko · 2023-02-18
**The paper describes existing and newly created (within this work) datasets for low-resource machine translation of Finno-Ugric languages and back-translation experiments, however it misses to answer one of the questions it raises (how much data is needed for low-resource back-translation).**

**Rating:** 6
**Confidence:** 5

**Review:**

The paper discusses parallel data collection for Finno-Ugric languages and experiments with back-translation.

The paper has the following contributions: 1) a new parallel dataset covering 20 languages, 2) translations of 250 sentences of the FLORES 200 dataset into nine languages, 3) experiments showing what translation directions to choose to back-translate data for to achieve higher low-resource translation direction quality in multilingual neural machine translation.

Issues:

1) The last sentence of the abstract lacks context. What are "carefully selected limited amounts of back-translation directions"? So far, the reader has no clue that the authors might (further) explore multilingual NMT. Therefore, the reader lacks context on how to understand this.

2) 118-119 - it is unclear how to understand the phrase "NMT model environments". What are "environments" in this context? Training toolkits, neural network architectures, machine translation paradigms/generations?

3) 123 - what are "universal models" in the context of neural machine translation? The referenced paper does not mention this term. This is also not a commonly known term in NMT. Therefore, please clarify what you mean by this!

4) 161-162 - it is false to assume/state that low-resource languages are rich in monolingual data. Consider rephrasing!

5) 252 - The "Tatoeba corpora" and "Tatoeba Translation Challenge" are not the same! The authors confuse data from the Tatoeba Translation Challenge with the Tatoeba corpus.

6) The experiments on back-translation ask a question (e.g., on line 100) that is not answered in their work. The authors want to know how much back-translated data is needed for low-resource NMT. But! What is low-resource NMT? How does one quantify what low-resource NMT is? What the authors identified was not how much is needed for (any/arbitrary) low-resource NMT. The authors even did not analyse amount of data, but rather what translation directions would data need to be back-translated to achieve a higher result on a particular data set! It would be interesting and beneficial for everyone to know the formula that allows to identify the needed amount for any dataset, but that is not what the authors identified. Therefore, I believe that they failed to answer the question they asked.

7) It is hard to follow the back-translation direction given the current wording in the paper. Does "L - rH" mean that data was back-translated from the low-resource languages into high-resource languages and not the other way around (high-resource into low-resource to help L->rH pairs)?

8) The link to the dataset nor its license is provided. The authors have also failed to specify whether the data will be available to anyone except themselves. This may have a negative effect on the work's impact since there is an abundance of publications on low-resource NMT (with back-translation) published, and the main contribution of the work is the data (and not the models and experiments on back-translation).

Language usage mistakes:

1) 147, 407, 417 - comma missing after the introductory phrase.

2) "article" -> "paper" or "publication"

3) 698 - My brain failed to parse the sentence! Consider rephrasing! 728 is another candidate with skipped words that hinder following the sentence.

In summary, I believe that the data the paper describes is very important and contributes substantially to low-resource machine translation in Finno-Ugric languages. However, I believe the authors got lost when performing the back-translation experiments. I believe that the paper would benefit if the authors would tone down a bit the contribution and the findings in this aspect. I.e., they performed experiments with exact data combinations, and on those combinations they got the presented results. They may indicate that there may be a tendency, but I believe that the experiments are hardly enough to make conclusions about necessary data amounts for back-translation for low-resource machine translation.


**Paper Type:**

Long paper

---

### Official Review · Reviewer_4F2x · 2023-03-08
**Towards MT for low-resource Finno-Ugric Languages**

**Rating:** 6
**Confidence:** 4

**Review:**

The paper reports ongoing work on developing neural machine translation for low-resource Finno-Ugric languages. The work includes collecting parallel and mono-lingual corpora, contributing data to the FLORES-200 dataset by adding data for new languages, and method development for the training of MT systems within the Finno-Ugric family of languages. In particular, the paper compares different strategies for back-translation in system development.

The paper reports the size of collected corpora (in thousands of sentences) and the amount of FLORES-data translated (250 for each language, out of 3000). Some errors and problems that have been found in the FLORES data are discussed in detail, but a more thorough discussion on the measures of quality ensurance that the project employs is wanting. For example, when it was clear that not all FLORES segments could be translated in time, a selection of segments based on assumed relevance for the Fenno-Ugric speakers could have been made, rather than just selecting rows from the beginning. Thus, segments that were found to be too long or contain alien concepts could have been discarded from the beginning.

The experiments use four different setups. They are reasonable although the figure 10% of available data seems arbitrary. As the smaller synthetic set often performs on a par with the larger (all-all) set, it could have been interesting to apply the same restriction also when the number of language pairs are reduced, as the settings L-H and L-rH generally are the ones that give the best results.

The paper points out a number of limitations of the study, the major one being that for many language pairs the available parallel data is exclusively or to a very high percentage based on the Bible. This was known from the start, however, and it would have been nice to see a discussion on what can be done about it.

I interpret the results as preliminary and possible to improve. The project is ambitious but still work in progress.The quality is fair and familiarity with relevant literature high. It is clear what has been done but not always why. I would rate the significance higher if the work is supported with data on what the respective language communities really ask for.



**Paper Type:**

Short paper

---

### Official Review · Reviewer_SwCZ · 2023-03-13
**Interesting Analysis of Backtranslation and MT for Low-resource Finno-Ugric Languages**

**Rating:** 8
**Confidence:** 3

**Review:**

This paper explores machine translation (MT) for low-resource Finno-Ugric languages. Specifically, it asks the research question of which languages should be involved in the training of a multilingual MT system with backtranslation. Surprisingly, the authors find that using a large set consisting of all languages is not the best option. Instead, subsets of (low-resource and high-resource) languages should be carefully chosen.

In addition, the authors create a 250-sentence version of FLORES-200 for the 9 languages used in their experiments, which will have value for research on low-resource MT for Finno-Ugric languages.

Additional comment: the captions of Table 3 could be more extensive, as the table was overall difficult to understand.

**Paper Type:**

Long paper

---

### Decision · Program_Chairs · 2023-03-17

Accept